# Predictive Approximate Bayesian Computation via Saddle Points

Yingxiang Yang[*]    Bo Dai[⋆]    Negar Kiyavash[†]    Niao He[*†]
{yyang172,kiyavash,niaohe} @illinois.edu
bohr.dai@gmail.com

## Abstract

Approximate Bayesian computation (ABC) is an important methodology for Bayesian inference when the likelihood function is intractable. Sampling-based ABC algorithms such as rejection- and K2-ABC are inefficient when the parameters have high dimensions, while the regression-based algorithms such as K- and DR-ABC are hard to scale. In this paper, we introduce an optimization-based ABC framework that addresses these deficiencies. Leveraging a generative model for posterior and joint distribution matching, we show that ABC can be framed as saddle point problems, whose objectives can be accessed directly with samples. We present *the predictive ABC algorithm (P-ABC)*, and provide a probabilistically approximately correct (PAC) bound for its learning consistency. Numerical experiment shows that P-ABC outperforms both K2- and DR-ABC significantly.

## 1 Introduction

Approximate Bayesian computation (ABC) is an important methodology to perform Bayesian inference on complex models where likelihood functions are intractable. It is typically used in large-scale systems where the generative mechanism can be simulated with high accuracy, but a closed form expression for the likelihood function is not available. Such problems arise routinely in modern applications including population genetics [Excoffier, 2009, Drovandi and Pettitt, 2011], ecology and evolution [Csilléry et al., 2012, Huelsenbeck et al., 2001, Drummond and Rambaut, 2007], state space models [Martin et al., 2014], and image analysis [Kulkarni et al., 2014].

Formally, ABC aims to estimate the posterior distribution $p(\theta|y) \propto p(y|\theta)\pi(\theta)$ where $\pi(\theta)$ is the prior and $p(y|\theta)$ is the likelihood function that represents the underlying model. The word "approximate" (or "A" in the abbreviation "ABC") refers to the fact that the joint distribution $\pi(\theta)p(y|\theta)$ is only available through fitting simulated data $\{(\theta_j, y_j)\}_{j=1}^N \sim p(y|\theta)\pi(\theta)$. Based on how the fitting is performed, existing ABC methods can be summarized into two main categories: sampling- and regression-based algorithms.

**Sampling-based algorithms.** A sampling-based algorithm directly approximates the likelihood function using simulated samples that are "close" to the true observations. This closeness between simulated samples $y_i$ and the true observation $y^*$ is measured by evaluating a similarity kernel $K_\epsilon(y_i, y^*)$. Informative summary statistics are often used to simplify this procedure when the dimension of $y^*$ is large, e.g., [Joyce and Marjoram, 2008, Nunes and Balding, 2010, Blum and François, 2010, Wegmann et al., 2009, Blum et al., 2013]. Representative algorithms in this category include rejection ABC, indirect score ABC [Gleim and Pigorsch], K2-ABC [Park et al., 2016], distribution regression ABC (DR-ABC) [Mitrovic et al., 2016], expectation propagation ABC (EP-ABC) [Barthelmé and Chopin, 2011], random forest ABC [Raynal et al., 2016], Wasserstein ABC [Bernton et al., 2017], copula ABC [Li et al., 2017], and ABC aided by neural network classifiers [Gutmann

---

[*]Department of Electrical and Computer Engineering, University of Illinois at Urbana-Champaign.
[†]Department of Industrial and Enterprise Systems Engineering, University of Illinois at Urbana-Champaign.
[⋆]Google Brain.

et al., 2014, 2016]. The aforementioned work can be viewed under a unified framework that approximates the posterior $p(\theta|y^*)$ using a weighted average of $p(y|\theta)\pi(\theta)$ over $\mathcal{Y}$ with the majority of the mass concentrated within a small region around $y^*$:

$$p_\epsilon(\theta|y^*) \propto \int_\mathcal{Y} K_\epsilon(s_y, s_{y^*})p(y|\theta)\pi(\theta)\mathrm{d}y \approx \sum_{i=1}^N \delta_{\theta_i} K_\epsilon(s_{y_i}, s_{y^*}). \qquad (1)$$

This weighted average is then approximated using the sample average with $\delta_{\theta_i} := \mathbb{1}[\theta = \theta_i]$ being the indicator function, and $s_y$ being the summary statistics for $y$. In other words, sampling-based algorithms reconstruct the posterior using a probability mass function where the mass distributes over the simulated $\theta_i \sim \pi(\theta_i)$, and, for each $\theta_i$, is proportional to the closeness of $y_i \sim p(y_i|\theta_i)$ and $y^*$. For example, when $K_\epsilon(s_{y_i}, s_{y^*}) = \mathbf{1}\{s_y = s_{y^*}\}$ with $s_y = y$, (1) recovers the true posterior asymptotically when the model parameter and the observation have discrete alphabets. When $K_\epsilon(s_y, s_{y^*}) = \mathbf{1}\{\rho(s_y, s_{y^*}) \leq \epsilon\}$ for some metric $\rho$, (1) reduces to rejection-ABC. When $K_\epsilon(s_y, s_{y^*}) = \exp(-\rho(s_y, s_{y^*})/\epsilon)$, (1) reduces to soft-ABC [Park et al., 2016], a variation of the synthetic likelihood inference [Wood, 2010, Price et al., 2018] under the Bayesian setting. Finally, when $K_\epsilon(s_y, s_{y^*})$ is the zero/one output of a neural network classifier, (1) reduces to ABC via classification [Gutmann et al., 2018].

Note that, most of the aforementioned algorithms require summary statistics and a smoothing kernel, which introduce bias, and suffer from information loss when the summary statistics are insufficient. To address the issue of having to select problem-specific summary statistics, Park et al. [2016] proposed K2-ABC, in which the summary statistics $s_{y^*}$ is replaced by the kernel embedding of the empirical conditional distributions of $p(y^*|\theta)$. When a characteristic kernel is selected, the kernel embedding of the distribution will be a sufficient statistics, and therefore does not incur information loss. Apart from directly choosing kernel embedding of the conditional distribution, other approaches exist to help reduce bias: for example, the recalibration technique proposed by Rodrigues et al. [2018]. However, despite their simplicity and continuous improvements, sampling-based ABC algorithms still suffer from the following deficiencies: (i) bias caused by the weighting kernel $K_\epsilon$, (ii) the potential need of large sample size when the dimensions of $\theta$ and $y^*$ are large, and (iii) the need to access the model every time a new observation is given.

**Regression-based algorithms.** Regression-based ABC algorithms establish regression relationships between the model parameter and the conditional distribution $p(y|\theta)$ within an appropriate function space $\mathcal{F}$. Representative algorithms in this category include high-dimensional ABC [Nott et al., 2014], kernel-ABC (K-ABC) [Blum et al., 2013], and distribution-regression-ABC (DR-ABC) [Mitrovic et al., 2016]. In DR-ABC, the kernel embeddings of the empirical version of the conditional distribution, $\{\mu_{\widehat{p}(y|\theta_i)}\}_{i=1}^N$, is first obtained from training data, and is then used to perform distribution regression:

$$h^* = \underset{h \in \mathcal{H}}{\mathrm{argmin}} \frac{1}{N} \sum_{i=1}^N \left| h(\mu_{\widehat{p}(y|\theta_i)}) - \theta_i \right|^2 + \lambda \|h\|_\mathcal{H}^2.$$

The algorithm then uses $h^*$ to predict the model parameter for any new set of data.

Contrary to the sampling-based algorithms, regression-based algorithms mitigate the bias introduced by the smoothing kernel. However, they do not provide an estimation for the posterior density. Meanwhile, it is often hard for such algorithms to scale. For example, the distribution regression involved in DR-ABC requires computing the inverse of an $N \times N$ kernel matrix, which has $\mathcal{O}(N^3)$ computation cost as the dataset scales.

Neither sampling- nor regression-based algorithms are satisfactory: while regression-based algorithms have better performances compared to the sampling-based algorithms, they are not scalable to high dimensions. Therefore, an important question is whether one can design an algorithm that can perform well on large datasets? In this paper, we propose an optimization-based ABC algorithm that can successfully address the deficiencies of both sampling- and regression-based algorithms. In particular, we show that ABC can be formulated under a unified optimization framework: *finding the saddle point of a minimax optimization problem*, which allows us to leverage powerful gradient-based optimization algorithms to solve ABC. More specifically, our contributions are three-fold:

- we show that the ABC problem can be formulated as a saddle point optimization through both joint distribution matching and posterior matching. This approach circumvents the

difficulties associated with choosing sufficient summary statistics or computing kernel matrices, as needed in K2- and DR-ABC. More critically, the saddle point objectives can be evaluated based purely on samples, without assuming any implicit form of the likelihood.

- we provide an efficient SGD-based algorithm for finding the saddle point, and provide a probabilistically approximately correct (PAC) bound guaranteeing the consistency of the solution to the problem.

- we compare the proposed algorithm to K2- and DR-ABC. The experiment shows that our algorithm outperforms K2- and DR-ABC significantly and is close to optimal on the toy example dataset.

## 2 Approximate Bayesian Computation via Saddle Point Formulations

When the likelihood function is given, the true posterior $p(\theta|y^*)$ given observation $y^*$ can be obtained by optimizing the *evidence lower bound* (ELBO) in the space $\mathcal{P}$ that contains all probability density functions [Zellner, 1988],

$$\min_{q(\theta) \in \mathcal{P}} \mathrm{KL}(q||\pi) - \mathbb{E}_{\theta \sim q}[\log p(y^*|\theta)], \qquad (2)$$

where KL denotes the Kullback-Leibler divergence: $\mathrm{KL}(q||\pi) = \mathbb{E}_{\theta \sim q}[\log \frac{q(\theta)}{\pi(\theta)}]$. When dealing with an intractable likelihood, this conventional optimization approach cannot work without combining it with methods that fit $p(y^*|\theta)$ with samples. In this paper, we introduce a new class of saddle point optimization objectives that allow the learner to directly leverage the samples from the likelihood $p(y^*|\theta)$, which is available under the ABC setting, for estimating the posterior. The method we propose does not merely find $\theta^* = \mathrm{argmax}_\theta \, p(\theta|y^*)$ for any given observation $y^*$, but rather finds the optimal $p(\theta|y)$ as a function of both $\theta$ and $y$, or a representation of $\theta$ generated from $p(\theta|y)$ using a transportation reparametrization $\theta = f(y, \xi)$ for any data $y$ (an idea inspired by Kingma and Welling [2013]). We introduce our method below.

### 2.1 Saddle Point Objectives

**Joint distribution matching.** Recall that $p(y|\theta)\pi(\theta) = p(\theta|y)p(y)$. A natural idea for estimating the posterior is to match the empirical joint distributions, given the availability of sampling from the product of the prior distribution and the model, $p(y|\theta)\pi(\theta)$, and from the product of the estimated posterior distribution and the marginal $q(\theta|y)p(y)$. Using an $f$-divergence associated with some convex function $\nu$, defined by $\mathsf{D}_\nu(p_1, p_2) = \int p_2(x)\nu\left(p_1(x)/p_2(x)\right) \mathrm{d}x$, as our loss function, we have the following divergence minimization problem for ABC:

$$p(\theta|y) = \underset{q(\theta|y) \in \mathcal{P}}{\mathrm{argmin}} \, \mathsf{D}_\nu\left(p(y|\theta)\pi(\theta), q(\theta|y)p(y)\right). \qquad (3)$$

This problem aims to recover the optimal posterior within the space of density functions $\mathcal{P}$. Ideally, if $\mathcal{P}$ is large enough such that $p(\theta|y) \in \mathcal{P}$, then (3) recovers the true posterior distribution.

However, the above optimization problem is still difficult to solve since $D_\nu$ is nonlinear with respect to $q(\theta|y)$. This nonlinearity makes gradient computation hard as the computation of the $f$-divergence still requires the value of $p(y|\theta)$, which is not available under the ABC setting, and cannot be computed directly through samples obtained from the joint distribution. In order to make the objective accessible through samples, we apply Fenchel duality and the interchangeability principle as introduced in Dai et al. [2017], which yield an equivalent saddle point reformulation. We state the detailed procedure in the following proposition.

**Proposition 1.** The divergence minimization (3) is equivalent to the following saddle point problem:

$$\min_{q(\theta|y) \in \mathcal{P}} \max_{u(\theta,y) \in \mathcal{U}} \Phi(f, u) := \mathbb{E}_{(\theta,y) \sim p(y|\theta)\pi(\theta)} \left[u(\theta, y)\right] - \mathbb{E}_{\theta \sim q(\theta|y), y \sim p(y)} \left[\nu^*\left(u(\theta, y)\right)\right], \qquad (4)$$

where $\mathcal{U}$ is a function space containing $u^*(\theta, y) = \nu'(\frac{p(y|\theta)p(\theta)}{p(\theta|y)p(y)})$ and $\nu^*$ is the Fenchel dual of $\nu$.

*Proof.* **Step 1**: By the definition of $f$-divergence,

$$\mathsf{D}_\nu(p(y|\theta)\pi(\theta)\|q(\theta|y)p(y)) = \mathbb{E}_{q(\theta|y)p(y)} \left[\nu\left(\frac{p(y|\theta)\pi(\theta)}{q(\theta|y)p(y)}\right)\right].$$

**Step 2**: Apply Fenchel duality $\nu(x) = \sup_u(ux - \nu^*(x))$ and obtain

$$D_\nu(p(y|\theta)\pi(\theta)\|q(\theta|y)p(y)) = \mathbb{E}_{q(\theta|y)p(y)}\left[\sup_u u \cdot \frac{p(y|\theta)\pi(\theta)}{q(\theta|y)p(y)} - \nu^*(u)\right].$$

**Step 3**: The interchangeability principle in Dai et al. [2017] suggests

$$D_\nu(p(y|\theta)\pi(\theta)\|q(\theta|y)p(y)) = \sup_{u\in\mathcal{U}} \mathbb{E}_{q(\theta|y)p(y)}\left[u(\theta,y) \cdot \frac{p(y|\theta)\pi(\theta)}{q(\theta|y)p(y)} - \nu^*(u(\theta,y))\right].$$

**Step 4**: By change of measure, we have (3) equivalent to (4). $\square$

The class of $f$-divergence covers many common divergences, including the KL divergence, Pearson $\chi^2$ divergence, Hellinger distance, and Jensen-Shannon divergence. Apart from $f$-divergences, we can also employ other metrics to measure the distance between $p(y|\theta)\pi(\theta)$ and $p(\theta|y)p(y)$, e.g., the Wasserstein distance. If the training data come with labels, we can also choose the objective function to be the mean square error between the label and the maximum a posterior estimate from $p(\theta|y)$. [2] From a density ratio estimation perspective, the optimal solution of the dual variable, $u(\theta,y)$, is a discriminator that distinguishes the true and estimated joint distributions by computing their density ratios, which is related to the ratio matching in Mohamed and Lakshminarayanan [2016].

**Posterior matching.** Another way to learn the posterior representation is by directly matching the posterior distributions. Similar to the objective function defined in K-ABC, we have

$$\min_{q(\theta|y)\in\mathcal{P}} \max_{h(\theta)\in\mathcal{H}} \mathbb{E}_y\left[(\mathbb{E}_{\theta|y}[h(\theta)] - \mathbb{E}_{\theta\sim q(\theta|y)}[h(\theta)])^2\right]. \tag{5}$$

Directly solving the optimization (5) is difficult due to the inner conditional expectation, but a saddle point formulation can be obtained by applying the same technique we used to obtain (4) (see Appendix C for detailed derivations):

$$\min_{\substack{q(\theta|y)\in\mathcal{P} \\ }} \max_{\substack{h(\theta)\in\mathcal{H} \\ v(y)\in\mathcal{V}}} \mathbb{E}_{(\theta,y)\sim p(y|\theta)\pi(\theta)}[v(y)h(\theta)] - \mathbb{E}_{(\theta,y)\sim q(\theta|y)p(y)}[v(y)h(\theta)] - \frac{1}{4}\mathbb{E}_y[v^2(y)] \tag{6}$$

where $\mathcal{V}$ is the entire space of functions on $\mathcal{Y}$. The resulting saddle point objective (6) is much easier to solve than (5) and stochastic gradient-based methods could be applied in particular.

### 2.2 Representations of $u(\theta,y)$ and $q(\theta|y)$

Under the most general setting where $\mathcal{P}$ and $\mathcal{U}$ are closed and bounded function spaces, the saddle point objective (4) is convex-concave. Practically, different representation methods can be used for $u(\theta,y)$ and $q(\theta|y)$, for which different optimization techniques can be applied to solving (4). Below, we discuss several commonly used options.

**Gaussian mixtures.** Consider the following Gaussian mixture representation for $q(\theta|y)$ and $u(\theta,y)$:

$$q(\theta|y) = \sum_{i=1}^m c_i^{(q)}(y) \cdot \mathcal{N}(\mu_i^{(q)}, \Sigma^{(q)}; \theta) \quad \text{and} \quad u(\theta,y) = \sum_{i=1}^m c_i^{(u)} \cdot \mathcal{N}(\mu_i^{(u)}, \Sigma^{(u)}; (\theta,y)). \tag{7}$$

The coefficients $c_1^{(u)}, \ldots, c_m^{(u)}$ are positive real numbers while $c_1^{(q)}(y), \ldots, c_m^{(q)}(y)$ are $y$-dependent coefficients. A simple way to guarantee that the summation of $c_i^{(q)}(y)$ is one for any $y$ is to assume that they take the form of softmax functions:

$$c_i^{(q)}(y) = \frac{\exp([1, y^\top] \cdot c_i^{(q)})}{\sum_{j=1}^m \exp([1, y^\top] \cdot c_j^{(q)})}, \quad \forall i \in \{1, \ldots, m\}, \tag{8}$$

with $c_1^{(q)} = 0$. This makes (4) convex for $c_i^{(q)}$ and concave for $c_i^{(u)}$.

**Reparametrization.** When the dimensions of $\theta$ and $y$ increase, the conditional distribution $q(\theta|y)$ quickly becomes difficult to represent using parametric models. An effective way to implicitly represent $q(\theta|y)$ is to use a sampler $f(\xi, y) \in \mathcal{F}$ for a function space $\mathcal{F}$, in which $\theta$ is sampled using $\theta = f(\xi, y)$ using a pre-determined distribution $\xi \sim p_0(\xi)$. This idea is inspired by the reparametrization technique used in variational autoencoders (VAEs) and neural networks. In our case, both $f$ and $u$ can be represented using functions in reproducing kernel Hilbert spaces (RKHSs) or neural networks.

## 2.3 Discussions

The saddle point framework is closely related to both regression- and GAN-based ABC algorithms.

**Relationship with regression-based ABC algorithms**. Regression-based ABC algorithms, such as K-ABC, aim to compute the conditional expectation of the posterior by finding its conditional kernel embedding $C(y) : \mathcal{Y} \to \mathcal{H}$ in an RKHS. With such parametrization, the objective (5) becomes

$$\min_{C:\mathcal{Y}\to\mathcal{H}} L(C) := \sup_{\|h\|_{\mathcal{H}}\leq 1} \mathbb{E}_y[(\mathbb{E}_{\theta|y}[h(\theta)] - \langle h, C(y)\rangle_{\mathcal{H}})^2].$$

This problem is further relaxed to a distribution regression problem by swapping the square operator with the inner expectation, which leads to minimizing $\mathbb{E}_{\theta,y}[\|K(\cdot,\theta) - C(y)\|^2]$, an upper bound of $L(C)$. Specifically, we have

$$\sup_{\|h\|_{\mathcal{H}}\leq 1} \mathbb{E}_y \left[ \left(\mathbb{E}_{\theta|y}\left[h(\theta)\right] - \langle h, C(y)\rangle_{\mathcal{H}}\right)^2 \right] \leq \sup_{\|h\|_{\mathcal{H}}\leq 1} \mathbb{E}_y \left[ \left(\mathbb{E}_{\theta|y}\left[\langle h, k(\cdot,\theta)\rangle\right] - \langle h, C(y)\rangle_{\mathcal{H}}\right)^2 \right]$$

$$\leq \sup_{\|h\|_{\mathcal{H}}\leq 1} \mathbb{E}_{\theta,y}\left[\langle h, k(\cdot,\theta) - C(y)\rangle^2\right] \leq \sup_{\|h\|_{\mathcal{H}}\leq 1} \|h\|_{\mathcal{H}}^2 \, \mathbb{E}_{\theta,y}\left[\|k(\cdot,\theta) - C(y)\|^2\right]$$

$$= \mathbb{E}_{\theta,Y}\left[\|k(\cdot,\theta) - C(y)\|^2\right].$$

In contrast, the proposed optimization framework for posterior matching does not restrict $h \in \mathcal{H}$. Moreover, the saddle point objective (6) is an exact reformulation of (5), rather than an upper bound.

**Relationship with GAN-based ABC algorithms.** GAN-based algorithms leverage the representation power of the neural networks to optimize the ELBO. One example is the use of variational autoencoder (VAE), where both $q$ and $p$ in (2) are represented by Gaussian distributions parameterized by neural networks. Better performances have been observed in Mescheder et al. [2017] by embedding the optimal value of $q(\theta|y)$ as the optimal solution of a real-valued discriminator network, equivalent to performing reparametrization. However, compared to the saddle point formulation, Mescheder et al. [2017] requires computing an additional layer of optimization due to the embedding performed. Meanwhile, when the underlying parameter is discrete, the saddle point formulation can be viewed as a special case of conditional GAN (CGAN) [Mirza and Osindero, 2014].

## 3 Algorithm and Theory

In this section, we introduce a concrete algorithm named *predictive-ABC* (P-ABC) that solves the finite-sample approximation (i.e., empirical risk) of the saddle point problem. For the sake of presentation, we consider the empirical risk of (4), where the empirical expectations are taken over $N$ samples $\{(\theta_i, y_i)\}_{i=1}^N$:

$$\min_{q(\theta|y)\in\mathcal{P}} \max_{u(\theta,y)\in\mathcal{U}} \widehat{\Phi}_N(q, u) := \widehat{\mathbb{E}}_{(\theta,y)\sim p(y|\theta)\pi(\theta)} u(\theta, y) - \widehat{\mathbb{E}}_{y\sim p(y)} \left\{ \mathbb{E}_{\theta\sim q(\theta|y)}[\nu^*(u(\theta, y))|y] \right\}. \quad (9)$$

We denote the optimal solution as $q_N^*$ and $u_N^*$. In the following, we first introduce a general form of P-ABC, followed by customizations to different representation methods for $q(\theta|y)$ and $u(\theta, y)$. We then derive a probabilistically approximately correct (PAC) learning bound on the statistical error

$$\epsilon_N = \mathsf{D}_\nu(p(y|\theta)\pi(\theta), q_N^*(\theta|y)p(y)) - \mathsf{D}_\nu(p(y|\theta)\pi(\theta), q^*(\theta|y)p(y)),$$

which holds for closed and bounded function spaces $\mathcal{P}$ and $\mathcal{U}$ in general, with $q^*$ and $u^*$ denoting the solution to (4). Lastly, we present the convergence results of P-ABC. For representations of $q$ and $u$ such that the objective function is convex-concave, e.g. Gaussian mixture representations, we present the convergence of Algorithm 1. For the representation using reparametrization and neural networks, the convergence behavior of P-ABC remains largely an open problem.

### 3.1 The P-ABC Algorithm

We introduce P-ABC for solving (9), the empirical counterpart of (4), in Algorithm 1. This algorithm, in its general form, performs iterative updates to $q$ and $u$ using first-order information. The computation of stochastic gradients under the representations presented in Section 2 can be found in Appendix A.

### 3.2 Theoretical Properties

**Learning bound.** By invoking the tail inequality in Antos et al. [2008] and the $\epsilon$-net argument, we have the following theorem, the proof of which can be found in Appendix B.

---

**Algorithm 1** Predictive ABC (P-ABC)

---

**Input:** maximum number of iterations $T$. Prior distribution $\pi(\theta)$, model $p(y|\theta)$. Step sizes $\{\eta_k^u\}_{k=1}^T$ and $\{\eta_k^q\}_{k=1}^T$, samples $\{(\theta_i, y_i)\}_{i=1}^N$, objective function $\widehat{\Phi}_N$.

**Initialize:** $q_1, u_1$.

**for** $k = 1$ **to** $T$ **do**

　　Randomly select $(\theta_k, y_k) \in \{(\theta_i, y_i)\}_{i=1}^N$, sample $\widetilde{\theta}_k \sim q_k(\theta|y_k)$.

　　Compute stochastic gradients of $\widehat{\Phi}_N(q_k, u_k)$, denoted by $\nabla_q \widehat{\phi}_k(q_k, u_k)$ and $\nabla_u \widehat{\phi}_k(q_k, u_k)$, using $(\theta_k, y_k, \widetilde{\theta}_k)$.

　　Update $q$: $q_{k+1} \leftarrow \mathrm{Proj}_{\mathcal{P}}(q_k - \eta_k^q \cdot \nabla_q \widehat{\phi}_k(q_k, u_k))$.

　　Update $u$: $u_{k+1} \leftarrow \mathrm{Proj}_{\mathcal{U}}(u_k + \eta_k^u \cdot \nabla_u \widehat{\phi}_k(q_k, u_k))$.

**end for**

**Output:** $\bar{q}_T = \frac{\sum_{k=1}^T \eta_k^q q_k}{\sum_{k=1}^T \eta_k^q}$, and $\bar{u}_T = \frac{\sum_{k=1}^T \eta_k^u u_k}{\sum_{k=1}^T \eta_k^u}$.

---

**Theorem 1.** Suppose $\{(\theta_i, y_i)\}_{i=1}^N$ is a $\beta$-mixing sequence [3] with $\beta_m \leq \bar{\beta} \exp(-bm^\kappa)$ for constants $\bar{\beta}$, $b$ and $\kappa$, and suppose that function class $\mathcal{U} \times \mathcal{P}$ has a finite pseudo dimension $D$. [4] In addition, suppose that $u \in [-C_u, C_u]$ and the Fenchel dual satisfies $\nu^*(u) \leq C_\nu$. Then, with probability $1 - \delta$,

$$\epsilon_N \leq \sqrt{\frac{C_1 (\max(C_1/b, 1))^{1/\kappa}}{C_2 N}},$$

where $C_1 = \log N^{\frac{D}{2}} e \delta^{-1} + [\log(2 \max(16e(D+1)C_2^{\frac{D}{2}}, \bar{\beta}))]_+$ and $C_2 = (512(C_\nu + C_u)^2)^{-1}$.

Theorem 1 applies to all the formulations we introduced in Section 2, for which learning is consistent at a rate of $\mathcal{O}(N^{-1/2} \log N)$, with $N$ being the number of samples, when the empirical saddle point approximation can be exactly solved. Below, we discuss the convergence of Algorithm 1.

**Convergence of P-ABC.** From a theoretical perspective, global convergence of first-order methods such as stochastic gradient descent (SGD) can be achieved when the objective function $\widehat{\Phi}_N$ is convex-concave. For example, when $u$ and $q$ are Gaussian mixtures or belong to RKHSs. More often than not, the objective function is not convex-concave, for which stochastic gradient descent (SGD) based algorithms are only guaranteed to converge towards a stationary point in certain restricted cases [Sinha et al., 2017, Li and Yuan, 2017, Kodali et al., 2018]. Below, we provide the convergence results for Algorithm 1 when $\widehat{\Phi}_N$ is convex-concave.

Consider the standard metric for evaluating the quality of any pair of estimates $\bar{q}_T$ and $\bar{u}_T$:

$$\varepsilon(\bar{q}_T, \bar{u}_T) = \max_{u \in \mathcal{U}} \widehat{\Phi}_N(\bar{q}_T, u) - \min_{q \in \mathcal{P}} \widehat{\Phi}_N(q, \bar{u}_T).$$

We have the following result (See Appendix D for proof).

**Theorem 2** (Convergence of P-ABC). Suppose that $\mathcal{P}$ and $\mathcal{U}$ are closed and bounded function spaces with diameters $D_{\mathcal{P}}$ and $D_{\mathcal{U}}$, respectively. Let $\widehat{\Phi}_N$ be convex-concave and $L_N$-Lipschitz. Then, for the outputs of Algorithm 1 with $T$ iterates and whose step sizes satisfy $\eta_k^q = \eta_k^u = \eta_k$, we have

$$\mathbb{E}\left[\varepsilon(\bar{q}_T, \bar{u}_T)\right] \leq \frac{D_{\mathcal{P}}^2 + D_{\mathcal{U}}^2 + \sum_{k=0}^T 2\eta_k^2 L_N^2}{2 \sum_{k=0}^T \eta_k}.$$

Theorem 2 applies to the cases when $\mathcal{P}$ and $\mathcal{U}$ are spaces for Gaussian mixture coefficients or RKHSs, in which case $\widehat{\Phi}_N$ is convex-concave. It suggests that if the sequence of step sizes satisfies $\sum_{k=1}^\infty \eta_k = \infty$ and $\sum_{k=1}^\infty \eta_k^2 < \infty$, then $\lim_{T \to \infty} \varepsilon(\bar{q}_T, \bar{u}_T) = 0$. In this case, we choose $\eta_k = \Theta(k^{-1/2}$. Together with Theorem 1, we know that the overall error, contributed by the summation of the learning error and the optimization error, can be bounded by $\mathcal{O}(N^{-1/2} \log N)$ upon selecting $T = \Theta(N)$.

---

　　[3]A discrete time stochastic process is mixing if widely separated events are asymptotically independent. Here, $\beta_m$ provides an upper bound on the dependency of two events separated by $n$ intervals of time. See Meir [2000] for a detailed definition.

　　[4]Pseudo dimension, also known as the Pollard dimension, is a generalization of VC dimension to the function class (see chapter 11 of Anthony and Bartlett [2009]).

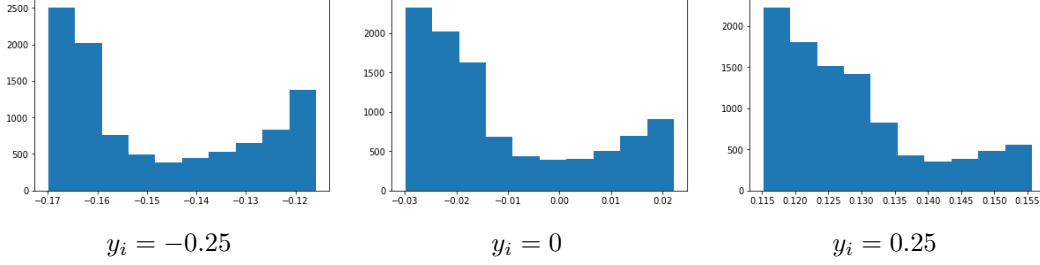

$$y_i = -0.25 \qquad\qquad y_i = 0 \qquad\qquad y_i = 0.25$$

Figure 1: Empirical distribution of $q(\theta|y_i)$ induced by the histogram of $f(y_i, \xi)$ computed from 1E4 i.i.d. samples of $\xi \sim p_0(\xi)$ for specific choices of $y_i$.

## 4    Numerical Experiment

We test the performance of P-ABC and compare the result with K2- and DR-ABC as representatives from sampling- and regression-based ABC algorithms.

### 4.1    Synthetic Dataset I: Superposition of Uniform Distributions

Consider $\theta \in \mathbb{R}^d$ and $\pi(\theta) = \mathbb{1}\{\theta \in [-0.5, 0.5]^d\}$. Let $p(y|\theta)$ be specified by $y = \theta + u$ with $u$ uniformly distributed over $[-0.5, 0.5]^d$. The training samples are $\{(\theta_i, \{y_{ij}\}_{j=1}^M)\}_{i=1}^N$ generated with independent $\theta_i$'s and $u_{ij}$'s. Denote $Y = \{Y_i\}_{i=1}^N$ with $Y_i = \{y_{ij}\}_{j=1}^M$, and for any $y$ and $\theta$, denote the $k$-th coordinate of them as $y[k]$ and $\theta[k]$, respectively. The posterior can then be written as

$$p(\theta_i|Y_i) \propto \prod_{k=1}^d \mathbb{1}\left[\max\{-0.5, \max_{j\in\{1,\dots,M\}} y_{ij}[k] - 0.5\} \leq \theta_i[k] \leq \min\{0.5, \min_{j\in\{1,\dots,M\}} y_{ij}[k] + 0.5\}\right],$$

which is a uniform distribution whose boundary on the $k$-th dimension is characterized by the values of the maximum and minimum values of the $k$-th coordinate among all $y_{ij}$'s. Due to the fact that K2- and DR-ABC evaluate their performances using the mean square error, we use predictive ABC (P-ABC) to find the optimal minimum mean square error (MMSE) estimator for $\theta_i$'s. We denote the optimal estimator for $\theta_i$ as $\widehat{\theta}_i^{\mathrm{opt}}$, which has a closed form solution with the $k$-th coordinate being

$$\widehat{\theta}_i^{\mathrm{opt}}[k] = \begin{cases} \frac{1}{2} \cdot \max_{j\in\{1,\dots,M\}} y_{ij}[k], & \min_{j\in\{1,\dots,M\}} y_{ij}[k] \geq 0 \\ \frac{1}{2} \cdot \min_{j\in\{1,\dots,M\}} y_{ij}[k], & \max_{j\in\{1,\dots,M\}} y_{ij}[k] \leq 0 \\ \frac{1}{2}\left(\max_{j\in\{1,\dots,M\}} y_{ij}[k] + \min_{j\in\{1,\dots,M\}} y_{ij}[k]\right), & \min_j y_{ij}[k] \leq 0 \leq \max_j y_{ij}[k] \end{cases},$$

for all $k \in \{1, \dots, d\}$. A sub-optimal estimator for this example is $\widehat{\theta}_i^{\mathrm{ave}} = M^{-1} \sum_{j=1}^M y_{ij}$, which exploits the information that the expectation of the noise is a zero vector. We include these two closed-form estimators in our benchmarks in addition to K2- and DR-ABC.

**The scalar case.** We first examine the case when $d = 1$ and $M = 1$. That is, when $\theta_i \in \mathbb{R}$ and when each $\theta_i$ only corresponds to one $y_i$. We compare the performance of P-ABC, obtained under the posterior matching objective with reparametrization representation of $q(\theta|y)$ and $u(\theta, y)$ with $f(y, \xi)$ and $u(\theta, y)$ using fully connected neural networks, to that of the theoretically optimal estimator, for which $\widehat{\theta}_i^{\mathrm{opt}} = y_i/2$. We train the neural networks using $N = 1000$ samples. Each neural network contains two fully connected layers of size 8 with exponential linear unit (ELU) activation functions, and the final output layer for $f$ is activated using the hyperbolic tangent. We choose $\xi \in \mathbb{R}$ and $p_0(\xi) \propto \mathbb{1}\{\xi \in [-1, 1]\}$, and use a learning rate of $10^{-4}$. In 2E5 iterations, P-ABC achieves 0.0413 mean square error (MSE) on the training set and 0.0416 MSE on the test set. The theoretically optimal MSE, obtained using $\widehat{\theta}_i^{\mathrm{opt}}$, is 0.0411 for the test set. Since $f$ does not directly show the posterior distribution, we plot a histogram in Figure 1 by evaluating $f(y, \xi)$ under 1E4 trials of $\xi$ for different values of $y$. We can see that the support of the empirical probability distribution is a very small interval near $y_i/2$, demonstrating that the output of P-ABC is nearly optimal. By comparison, since there is only one observation available, K2- and DR-ABC do not output meaningful results as the computation of the maximum mean discrepancy (MMD) statistics requires at least two observations. Lastly, for other choices of $M$ and $d$, the training and testing errors are reported in Table 1. The result shows that the performance of P-ABC was closer to the theoretical optimum than DR- and K2-ABC.

**Performance under higher dimensions.** We examine the performance of P-ABC when the dimension of the model parameter was higher. For illustration purpose, we choose $d \in \{1, 16, 128, 256\}$,

| MSE | P-ABC [test,train] | K2-ABC | DR-ABC | $\widehat{\theta}_{\text{opt}}$ | $\widehat{\theta}_{\text{ave}}$ |
|---|---|---|---|---|---|
| $d=1$ | [0.009,0.010] | 0.011 | 0.083 | 0.003 | 0.008 |
| $d=16$ | [0.182, 0.155] | 1.283 | 1.143 | 0.050 | 0.134 |
| $d=128$ | [2.749,1.793] | 21.478 | 10.730 | 0.409 | 1.064 |
| $d=256$ | [4.266,1.399] | 41.830 | 21.324 | 0.818 | 2.119 |

Table 1: MSE for estimating the model parameter with different dimensions using K2-, DR- and P-ABC. For K2- and DR-ABC, we set $\epsilon = 0.01$ when computing MMD. For P-ABC, the hidden layer sizes are 8,32,128,256 for different values of $d$, and the dimension of $\xi$ is 1,4,4,4, respectively.

and we assumed $M = 10$, i.e., $Y_i$ contains 10 samples for each parameter value. Once again, we use neural networks to represent $f$ and $u$ in P-ABC, for which we train with $N = 1000$ sets of samples.

To reduce the input dimension of the neural networks, for each input set of samples $Y_i$, we set $f(Y_i, \xi) = \frac{1}{M} \sum_{j=1}^{M} f(y_{ij}, \xi)$ and $u(\theta, Y_i) = \frac{1}{M} \sum_{i=1}^{M} u(\theta, y_{ij})$. More specifically, rather than taking the entire set of samples as the input, the neural network representing $f$ took each sample individually, and used their average as the final value of $f(Y_i, \xi)$. Under this setting, for 2E5 iterations, the obtained results are shown in Table 1. We can see that P-ABC outperformed both K2- and DR-ABC in all four cases, and when the dimension of $\theta_i$ was small, the performance of P-ABC was close to that of $\widehat{\theta}_i^{\text{ave}}$.

## 4.2 Synthetic Dataset II: Gaussian Mixtures

Consider a model where $\theta \in \mathbb{R}$ and $\pi(\theta) = \mathbb{1}\{\theta \in [-0.5, 0.5]\}$. The likelihood function $p(y|\theta)$ is characterized by a Gaussian mixture model: $y = (0.5 + \theta)\mathcal{N}(-1, 1) + (0.5 - \theta)\mathcal{N}(1, 1)$. In this example, we compare the performances between K2-, DR-, EP-, and the proposed P-ABC. For P-ABC, we adopt the same network structures for the neural networks representing $f$ and $u$ as in the previous example, and train them with $N = 4000$ sets of samples. Each set of samples contained $M = 250$ samples corresponding to the same model parameter. The same setting is used for evaluating the benchmarks. For the result, P-ABC achieves an MSE of 0.004, and EP-ABC achieves an MSE of 0.06. [5] During the implementation, we noticed that the EP-ABC requires Cholesky factorization for each iteration, which is computationally expensive and particularly sensitive to initialization. In fact, the run time of EP-ABC (10 sets of samples per minute) was significantly longer than P-ABC (200 sets of samples per minute). K2- and DR-ABC, by comparison, were unable to produce results for 100 sets of samples within 1 hour. This experiment demonstrated the efficiency of implementing the P-ABC algorithm.

**Discussions.** Although P-ABC demonstrates superior numerical performances over the benchmarks, it suffers from some of the deficiencies of the other existing ABC algorithms. One such deficiency is that the algorithm is prone to mismatched priors. To see this, we plotted the histogram of $f(y, \xi)$ with $y$ sampled from the model with the model parameter being 0, $\xi$ sampled from a uniform distribution over $[-1, 1]$, and with P-ABC trained on a mismatched prior. We skew the prior by substituting $\{\theta_i, y_i\}_{i=1}^{N}$ in Algorithm 1 with $\{\bar{\theta}_i, \bar{y}_i\}_{i=1}^{N}$ where $\bar{\theta} = (\theta + a)/(2a + 1)$ and $\bar{y}_i$ is the output from the model under $\bar{\theta}_i$. This transformation introduces bias between the true prior and the prior used for training, and, as can be seen in Figure 2, the range of the estimated parameter by P-ABC shifts away from true model parameter as the value of $a$ increases.

## 4.3 Ecological Dynamic System

Time series observations are an important application scenario for ABC. In this experiment, we compare the performances of K2-, DR- and P-ABC over the example of an ecological dynamic system studied in previous literatures (see Park et al. [2016] for example). The population dynamics follow the relationship

$$y_{t+1} = P y_{t-\tau} \exp\left(-\frac{y_{t-\tau}}{y_0}\right) e_t + y_t \exp(-\delta\epsilon_t),$$

an evolution dynamics parametrized by a 5-dimensional vector $\theta = (P, y_0, \sigma_d^2, \sigma_p^2, \tau, \delta) \in \mathbb{R}_+^5$. Let $Y = (y_1, \ldots, y_t)$ denote the set of samples that contains the population size data up to time $t$, the

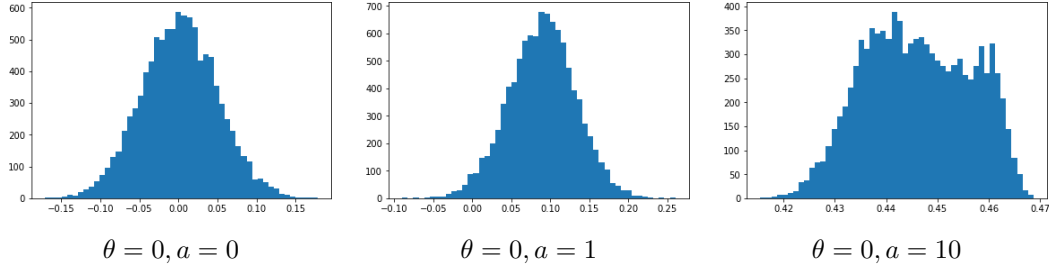

$\theta = 0, a = 0$          $\theta = 0, a = 1$          $\theta = 0, a = 10$

Figure 2: Impact of improper prior on P-ABC. Consider finding uniformly distributed $\theta \sim \mathcal{U}[-0.5, 0.5]$ from $y = (0.5 + \theta)\mathcal{N}(-1, 1) + (0.5 - \theta)\mathcal{N}(-1, 1)$. Improper priors are obtained by $\widetilde{\theta} = (\theta + a)/(2a + 1)$ with $a = 1, 10$. We see that training on improper prior injects bias into the output of P-ABC.

noise $e_t \sim \Gamma(\sigma_p^{-2}, \sigma_p^2)$, $\epsilon_t \sim \Gamma(\sigma_d^{-2}, \sigma_d^2)$. We sample each dimension of $\log \theta$ from a uniform distribution on $[-5, 2]$, and set $\tau = \lceil \tau \rceil$.

For P-ABC, we implemented a recurrent neural network (RNN) with long short-term memory (LSTM) cells to capture the dynamics of the underlying time series. The output of the LSTM cell is then plugged into a fully connected layer along with $\theta$ or $\xi$. The structures of the neural networks representing $f$ and $u$ are shown in Figure 4 in Appendix E.

When training P-ABC and the benchmarks, we set $t = 30$, and use $N = 1000$ sets of samples. For P-ABC, we set $\xi \in \mathbb{R}^4$, the size of thee LSTM cells to be 32, and the size of the fully connected layer to be 16. For K2- and DR-ABC, the samples within $Y$ were regarded as i.i.d.. The obtained result is shown in Figure 3, with the vertical axis denoting the MSE of the estimated parameter. We can see that P-ABC outperforms K2-ABC and DR-ABC on all aspects: the MSE was 12.9 for P-ABC, 24.7 for K2-ABC, and 16.4 for DR-ABC. In addition, P-ABC had the lowest average, quartile, and better performance on outliers.

## 5   Conclusion

In this paper, we presented a unifying optimization framework for ABC, named Predictive-ABC, under which we showed that ABC can be formulated as a saddle point problem for different objective functions. We presented a high-probability error bound that decays at the speed of $\mathcal{O}(N^{-1/2} \log N)$ with $N$ being the number of samples and we presented a stochastic-gradient-descent-based algorithm, P-ABC, to find the solution. In practice, P-ABC significantly outperforms K2- and DR-ABC, representatives for the state-of-the-art sampling- and regression-based algorithms, respectively.

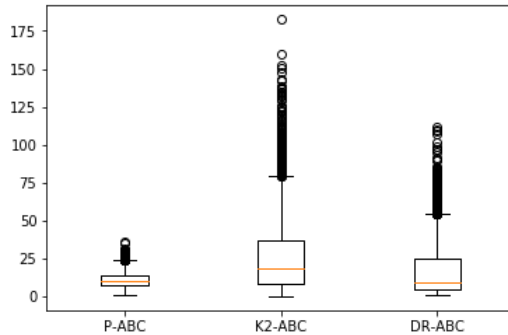

Figure 3: Statistics of MSEs for P-, K2- and DR-ABC trained on 1000 sequences of length 30.

## Acknowledgement

This work was supported in part by MURI grant ARMY W911NF-15-1-0479, ONR grant W911NF-15-1-0479, NSF CCF-1755829 and NSF CMMI-1761699.

## Footnotes

[2]Table 2 in Appendix provides some examples of divergences and the derivation of their corresponding saddle point objectives.

[5] Per implementation of the code made available online by Barthelmé and Chopin [2011].

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
