[Supplementary Material]

# Appendix

## A Computation of Stochastic Gradients for P-ABC

In this section, we introduce the computation of stochastic gradients under the representations specified in Section 2. In practice, the stochastic gradients are often computed using a mini-batch of samples. To avoid further introduction of new notations, we specify the computation of the full gradients for $\widehat{\Phi}_N$ which easily yields the form of stochastic gradients by substituting the empirical expectations over the entire set of $N$ samples with the empirical expectations over the minibatch samples, or $(\theta_k, y_k, \widetilde{\theta}_k)$ in the $k$-th iteration for Algorithm 1 when the minibatch size is 1.

We note that for any $q \in \mathcal{P}$ such that the integral can be interchanged with the gradient operator, $\nabla_q \mathbb{E}_q[f(X)] = \mathbb{E}_q[f(X)\nabla \log q(X)]$. This allows us to take gradient with respect to $q$ and use stochastic gradients for (9) assuming that a set of synthetic samples $\{\widetilde{\theta}_i\}_{i=1}^N$ can be drawn from $q(\theta|y)$ given $\{y_i\}_{i=1}^N$.

**Gaussian mixtures.** Denote the coefficient vectors for $q(\theta|y)$ and $u(\theta, y)$ in (7) and (8) as $\mathbf{c}^{(q)}$ and $\mathbf{c}^{(u)}$. Then, the gradients of $\widehat{\Phi}_N(q, u)$ with respect to $q(\theta|y)$ and $u(\theta, y)$ reduce to its gradients with respect to $\mathbf{c}^{(q)}$ and $\mathbf{c}^{(u)}$:

$$
\nabla_{\mathbf{c}^{(q)}} \widehat{\Phi}_N = -\widehat{\mathbb{E}}_{y\sim p(y)} \left\{ \mathbb{E}_{\theta\sim q(\theta|y)} \left[ \frac{\nu^*(u(\theta,y))}{q(\theta,y)} \cdot \begin{bmatrix} \nabla_{c_1^{(q)}} q(\theta|y) \\ \vdots \\ \nabla_{c_m^{(q)}} q(\theta|y) \end{bmatrix} \right] \right\},
$$

$$
\nabla_{\mathbf{c}^{(u)}} \widehat{\Phi}_N = \widehat{\mathbb{E}}_{p(y|\theta)\pi(\theta)}[\mathcal{N}(\boldsymbol{\mu}^{(u)}, \Sigma^{(u)}; (\theta, y))] - \widehat{\mathbb{E}}_{q(\theta|y)p(y)} \left[ \frac{\mathrm{d}\nu^*(u(\theta,y))}{\mathrm{d}u(\theta,y)} \cdot \mathcal{N}(\boldsymbol{\mu}^{(u)}, \Sigma^{(u)}; (\theta, y)) \right].
$$

where $\mathcal{N}(\boldsymbol{\mu}^{(u)}, \Sigma^{(u)}; (\theta, y))$ denotes a vector containing the values of Gaussian probability density functions whose means are specified in (7) and evaluated at $(\theta, y)$, while

$$
\nabla_{c_i^{(q)}} q(\theta|y) = \sum_{i=1}^m \frac{[1; y] \cdot \exp([1, y^\top] \cdot c_i^{(q)}) \cdot \sum_{j\neq i} \exp([1, y^\top] \cdot c_j^{(q)})}{\left( \sum_{j=1}^m \exp([1, y^\top] \cdot c_j^{(q)}) \right)^2} \cdot \mathcal{N}(\mu_i^{(q)}, \Sigma^{(q)}; \theta).
$$

**Reparametrization.** Consider the reparametrization $\theta = f(y, \xi)$ for $\theta \sim p(\theta|y)$. The gradients of $\widehat{\Phi}_N(f, u)$ in (9) can be computed from the chain rule:

$$
\nabla_f \widehat{\Phi}_N = -\widehat{\mathbb{E}}_{(y,\xi)\sim p(y)p_0(\xi)} \left\{ \frac{\mathrm{d}\nu^*(u(f(y,\xi),y))}{\mathrm{d}u(f(y,\xi),y)} \cdot \frac{\partial u(f(y,\xi),y)}{\partial f(y,\xi)} \cdot \nabla_f f(y,\xi) \right\},
$$

$$
\nabla_u \widehat{\Phi}_N = \widehat{\mathbb{E}}_{(\theta,y)\sim p(y|\theta)\pi(\theta)} \nabla_u u(\theta, y) - \widehat{\mathbb{E}}_{(y,\xi)\sim p(y)p_0(\xi)} \left[ \frac{\mathrm{d}\nu^*(u(f(y,\xi),y))}{\mathrm{d}u(f(y,\xi),y)} \cdot \nabla_u u(f(y,\xi),y) \right].
$$

When $\mathcal{F}$ and $\mathcal{U}$ are RKHSs, we have $\nabla_f f(y, \xi) = K_{\mathcal{F}}((y, \xi), \cdot)$ and $\nabla_u u(\theta, y) = K_{\mathcal{U}}((\theta, y), \cdot)$ with $K_{\mathcal{F}}$ and $K_{\mathcal{U}}$ being the reproducing kernels of $\mathcal{F}$ and $\mathcal{U}$, respectively, while $\Pi_{\mathcal{F}}$ and $\Pi_{\mathcal{U}}$ denote the projections onto $\mathcal{F}$ and $\mathcal{U}$. When $f$ and $u$ are represented by neural networks, $\nabla_f$ and $\nabla_u$ are the gradients with respect to the coefficients representing those neural networks, which can be efficiently calculated through back propagation.

## B Proof of Theorem 1

In this proof, we find the concentration bound for the statistical error for (4), which is defined as

$$
\epsilon_N = \mathsf{D}_\nu(p(y|\theta)\pi(\theta), q_N^*(\theta|y)p(y)) - \mathsf{D}_\nu(p(y|\theta)\pi(\theta), q^*(\theta|y)p(y)),
$$

where $q_N^*(\theta|y)$ is the distribution that optimizes (9) (for reparametrization $q_N^*(\theta|y)$ is the distribution induced by the optimal empirical solution $f_N^* \in \mathcal{F}$ with $N$ samples in (4)), and $q^*(\theta|y)$ is the solution

to (4). Let $\widehat{u} := \operatorname{argmax}_{u \in \mathcal{U}} \Phi(f_N^*, u)$, we can bound the learning error $\epsilon_N$ by

$$\epsilon_N = \max_{u \in \mathcal{U}} \Phi(q_N^*, u) - \max_{u \in \mathcal{U}} \Phi(q^*, u) = \Phi(q_N^*, \widehat{u}) - \Phi(q^*, u^*)$$
$$= \Phi(q_N^*, \widehat{u}) - \Phi(q^*, \widehat{u}) + \Phi(q^*, \widehat{u}) - \Phi(q^*, u^*)$$
$$\leq \Phi(q_N^*, \widehat{u}) - \Phi(q^*, \widehat{u}) \leq 2 \sup_{q \in \mathcal{P}, u \in \mathcal{U}} |\widehat{\Phi}_N(q, u) - \Phi(q, u)|.$$

In the following, we provide a high probability upper bound for $\sup_{q,u} |\widehat{\Phi}_N(q, u) - \Phi(q, u)|$.

## B.1 Technical Lemmas

The concentration bound requires Lemma 5 in Antos et al. [2008]:

**Lemma 1** (Lemma 5 [Antos et al., 2008]). Suppose that $Z_1, \ldots, Z_N \in \mathcal{Z}$ is a sequence that is stationary and $\beta$-mixing, and $\mathcal{G}$ is a class of bounded functions, then

$$\mathbb{P}\left(\sup_{g \in \mathcal{G}} \left| \frac{1}{N} \sum_{i=1}^N g(Z_i) - \mathbb{E}[g(Z_1)] \right| > \varepsilon \right) \leq 16 \mathbb{E}\left[ \mathcal{N}\left(\varepsilon/8, \mathcal{G}, (Z_i'; i \in H)\right)\right] \exp\left( \frac{-N\varepsilon^2}{256C^2} \right) +$$
$$+ 2m_N \beta_{k_N+1}$$

where $Z_i'$ is the "ghost" sample that mirrors $Z_i$, $\mathcal{N}(\epsilon/8, \mathcal{G}, (Z_i'; i \in H))$ is the covering number for $\mathcal{G}$, and $H = \cup_{j=1}^{m_N} H_i$ is the union of blocks in the sampling path.

The covering number in the above lemma can be bounded using the following result:

**Lemma 2** (Corollary 3 [Haussler, 1995]). For any set $\mathcal{X}$, and $x_1, \ldots, x_n \in \mathcal{X}$, assume $\mathcal{F}$ of functions on $\mathcal{X}$ are bounded within $[0, C]$ with pseudo-dimension $D_{\mathcal{F}} < \infty$. Then, for any $\varepsilon$,

$$\mathcal{N}(\varepsilon, \mathcal{F}, (x_1, \ldots, x_n)) \leq e(D_{\mathcal{F}} + 1) \left( \frac{2eC}{\varepsilon} \right)^{D_{\mathcal{F}}},$$

where $e$ is the Euler's number.

With the above two lemmas, we are ready to prove the theorem.

## B.2 Learning Error

For any sample $(\theta, y)$, let $\phi(q, u) = u(\theta, y) - \mathbb{E}_{\theta \sim q(\theta|y)}[\nu^*(u(\theta, y))|y]$. Then, the objective function in (4) can be written as $\Phi(q, u) = \mathbb{E}[\phi(q, u)]$, where the expectation is with respect to the joint distribution of $(\theta, y) \sim p(y|\theta)\pi(\theta)$.

By assumption, since $|u(\theta, y)| \leq C_u$, [6] the Fenchel dual in (4) can be bounded. In particular, we denote the upper bound as $\nu^*(u) = \sup_{u' \in [-C_u, C_u]} \langle u', u \rangle - \nu(u') \leq C_\nu$, [7] and hence

$$\phi(q, u) \leq C_\nu + C_u.$$

Invoking Lemma 1, in which we let $\mathcal{G} = \mathcal{P} \times \mathcal{U}$, we have

$$\mathbb{P}\left( \sup_{(q,u) \in \mathcal{P} \times \mathcal{U}} \left| \widehat{\Phi}_N(q, u) - \Phi(q, u) \right| > \epsilon \right) \leq 2m_N \beta_{k_N+1} +$$
$$+ 16 \mathbb{E}\left[ \mathcal{N}\left( \frac{\epsilon}{8}, \mathcal{P} \times \mathcal{U}, ((\theta_i, y_i), i \in H) \right) \right] \exp\left( \frac{-m_N \epsilon^2}{128(C_\nu + C_u)^2} \right).$$

Next, invoking Lemma 2 and by assumption, [8] we have

$$\mathbb{P}\left( \sup_{(q,u) \in \mathcal{P} \times \mathcal{U}} \left| \widehat{\Phi}_N(q, u) - \Phi(q, u) \right| > \epsilon \right) \leq 16 e(D + 1) \left( \frac{4eC}{\epsilon} \right)^D \exp\left( \frac{-m_N \epsilon^2}{128(C_\nu + C_u)^2} \right) +$$
$$+ 2m_N \beta_{k_N+1}.$$

Lastly, by setting $k_N = (N\epsilon^2 C_2/b)^{1/(\kappa+1)}$, with $C_2 = 1/(512(C_\nu + C_u)^2)$ and $m_N = N/2k_N$, we have the right-hand side of the above equation bounded by $\delta$ with

$$\epsilon = \sqrt{\frac{C_1(\max(C_1/b, 1))^{1/\kappa}}{C_2 N}},$$

and $C_1 = 0.5D \log N + \log(e/\delta) + [\log(2\max(16e(D+1)C_2^{D/2}, \bar{\beta}))]_+$, with $\bar{\beta}$ is such that $\beta_m \le \bar{\beta} \exp(-bm^\kappa)$.

## C  Derivation of objective functions

| Divergence | Saddle point objective |
|---|---|
| $\chi^2$ divergence | $\mathbb{E}_{(\theta,y)\sim p(y\mid\theta)\pi(\theta)}[u(\theta,y)] - \mathbb{E}_{\theta\sim f(y,\xi),\xi\sim p_0(\xi),y\sim p(y)}[u(\theta,y) + u^2(\theta,y)/4]$ |
| Wasserstein distance | $\mathbb{E}_{(\theta,y)\sim p(y\mid\theta)\pi(\theta)}[u(\theta,y)] - \mathbb{E}_{\theta\sim f(y,\xi),\xi\sim p_0(\xi),y\sim p(y)}[u(\theta,y)]$ |
| KL divergence | $\mathbb{E}_{(\theta,y)\sim p(y\mid\theta)\pi(\theta)}[u(\theta,y)] - \mathbb{E}_{\theta\sim f(y,\xi),\xi\sim p_0(\xi),y\sim p(y)}[1 + \log(u(\theta,y))]$ |

Table 2: A list of divergences and their corresponding saddle point objective.

In this section, we present the detailed derivation of the objective functions in Section 2 and give examples for deriving (4) for several choices of $f$-divergences in Table 2.

**Posterior Distribution Matching**   For the posterior distribution matching objective, we want to minimize

$$\min_{f\in\mathcal{F}} \max_{h\in\mathcal{H}} \mathbb{E}_y \left[ \left( \mathbb{E}_{\theta\mid y}[h(\theta)] - \mathbb{E}_{\theta\sim p_0(\xi)}[h(f(y,\xi))] \right)^2 \right].$$

By exploiting the dual embedding technique and the Fenchel duality, we have

$$\max_{h\in\mathcal{H}} \mathbb{E}_y \left[ \left( \mathbb{E}_{\theta\mid y}[h(\theta)] - \mathbb{E}_{\theta\sim p_0(\xi)}[h(f(y,\xi))] \right)^2 \right]$$

$$= \max_{h\in\mathcal{H}} \mathbb{E}_y \left[ \max_{v_y\in\mathbb{R}} v_y \left( \mathbb{E}_{\theta\mid y}[h(\theta)] - \mathbb{E}_{\xi\sim p_0(\xi)}[h(f(y,\xi))] - \frac{1}{2}v_y^2 \right) \right]$$

$$= \max_{h\in\mathcal{H}} \max_{v\in\mathcal{V}} \mathbb{E}_y \left[ v(y) \cdot \left( \mathbb{E}_{\theta\mid y}[h(\theta)] - \mathbb{E}_{\xi\sim p(\xi)}[h(f(y,\xi))] \right) - \frac{1}{2}v^2(y) \right],$$

thus achieving the equivalence between (5) and (6).

**Joint Distribution Matching**

- $\chi^2$**-divergence.** Recall that the divergence minimization objective (3) can be written in the saddle point formulation:

$$\min_{f\in\mathcal{F}} \max_{u\in\mathcal{U}} \mathbb{E}_{(\theta,y)\sim p(y\mid\theta)\pi(\theta)} [u(\theta,y)] - \mathbb{E}_{\theta\sim f(y,\xi),\xi\sim p_0(\xi),y\sim p(y)} [\nu^*(u(\theta,y))].$$

  For $\chi^2$ divergence, we have

$$\nu(x) = (x-1)^2.$$

  Therefore,

$$\nu^*(x) = \sup_y(\langle x,y\rangle - \nu(y)) = \sup_y(xy - (y-1)^2) = \frac{x^2 + 4x + 8}{4}.$$

  Plugging in $x = u(\theta,y)$ gives the result of $\nu^*(u(\theta,y))$, and the expression for $\Phi$ follows immediately.

$$\nu^*(x) = \sup_y(\langle x,y\rangle - \nu(y)) = \sup_y(\langle x,y\rangle - (y-1)^2) = x - \frac{x^2}{4}.$$

  Plugging in $x = u(\theta,y)$ gives the expression of $\nu^*(u(\theta,y))$, and hence we arrive at the conclusion.

- **KL divergence.** For KL divergence, $\nu(x) = \log x$. We thus have

$$\nu^*(x) = \sup_y(\langle x,y\rangle - \log y) = 1 + \log x.$$

  Therefore, letting $x = u(\theta,y)$ gives the expression of $\nu^*(u(\theta,y))$, and hence we arrive at the conclusion.

# D Proof of Theorem 2

Following the notations in Algorithm 1, we denote the updates of $q$ and $u$ at iteration $k$ by $q_k$ and $u_k$, respectively. Different representations of $q$ and $u$ will determine the detailed forms of $q_k$ and $u_k$. For example, when represented by Gaussian mixtures, both $u_k$ and $q_k$ in Algorithm 1 will be a coefficient vector. However, for the purpose of proving convergence, using an abstract form $u_k$ and $q_k$ is sufficient barring that $\mathcal{U}$ and $\mathcal{P}$ are closed and bounded.

By convex-concavity of the empirical loss function, we have

$$\widehat{\Phi}_N(q_k, u_k) - \widehat{\Phi}_N(q, u_k) \leq \langle \nabla_q \widehat{\Phi}_N(q_k, u_k), q_k - q \rangle,$$

and

$$\widehat{\Phi}_N(q_k, u) - \widehat{\Phi}_N(q_k, u_k) \leq \langle \nabla_u \widehat{\Phi}_N(q_k, u_k), u - u_k \rangle$$

for any $q \in \mathcal{P}$ and $u \in \mathcal{U}$. Combining these two inequalities gives

$$\widehat{\Phi}_N(q_k, u) - \widehat{\Phi}_N(q, u_k) \leq \langle \nabla_q \widehat{\Phi}_N(q_k, u_k), q_k - q \rangle + \langle \nabla_u \widehat{\Phi}_N(q_k, u_k), u - u_k \rangle.$$

It is worth clarifying, at this point, that the gradient symbol we used for $\widehat{\Phi}_N$ so far refer to the actual gradient rather than the stochastic gradients given in Section 3.1. However, they are closely related by the fact that the expectation of the stochastic gradient is the gradient. To avoid confusion, we use $\widehat{\nabla}_q \widehat{\Phi}_N$ and $\widehat{\nabla}_u \widehat{\Phi}_N$ to represent the stochastic gradients, for which we have

$$\mathbb{E}\left[\widehat{\nabla}_u \widehat{\Phi}_N(q, u)\right] = \nabla_u \widehat{\Phi}_N(q, u) \qquad \text{and} \qquad \mathbb{E}\left[\widehat{\nabla}_q \widehat{\Phi}_N(q, u)\right] = \nabla_q \widehat{\Phi}_N(q, u),$$

where the expectation is taken over the the second term in (9), where we have used $\theta \sim q(\theta|y)$ to derive the stochastic gradients.

By convexity of $\widehat{\Phi}_N$, we have

$$\varepsilon(\bar{q}_T, \bar{u}_T) = \max_{u \in \mathcal{U}} \widehat{\Phi}_N(\bar{q}_T, u) - \min_{q \in \mathcal{P}} \widehat{\Phi}_N(q, \bar{u}_T)$$

$$= \max_{u \in \mathcal{U}} \widehat{\Phi}_N\left(\frac{\sum_{k=1}^{T} \eta_k q_k}{\sum_{k=1}^{T} \eta_k}, u\right) - \min_{q \in \mathcal{P}} \widehat{\Phi}_N\left(q, \frac{\sum_{k=1}^{T} \eta_k u_k}{\sum_{k=1}^{T} \eta_k}\right)$$

$$\leq \max_{u \in \mathcal{U}} \frac{\sum_{k=1}^{T} \eta_k \widehat{\Phi}_N(q_k, u)}{\sum_{k=1}^{T} \eta_k} - \min_{q \in \mathcal{P}} \frac{\sum_{k=1}^{T} \eta_k \widehat{\Phi}_N(q, u_k)}{\sum_{k=1}^{N} \eta_k}$$

$$\leq \frac{\max_{u \in \mathcal{U}, q \in \mathcal{P}} \left\{ \sum_{k=0}^{T} \left( \eta_k \langle \nabla_q \widehat{\Phi}_N(q_k, u_k), q_k - q \rangle - \eta_k \langle \nabla_u \widehat{\Phi}_N(q_k, u_k), u_k - u \rangle \right) \right\}}{\sum_{k=1}^{T} \eta_k}.$$

We now prove that the expectation of the numerator is upper bounded by the numerator of the right-hand side of the bound in the statement of Theorem 2, which will bring us to the conclusion.

To prove this, we first note that, by the contractivity of the projection operator, we have, for any $q \in \mathcal{P}$,

$$\mathbb{E}\|q_{k+1} - q\|_{\mathcal{P}}^2 = \mathbb{E}\left\|\Pi_{\mathcal{P}}(q_k - \eta_k \widehat{\nabla}_q \widehat{\Phi}_N(q_k, u_k)) - \Pi_{\mathcal{P}}(q)\right\|_{\mathcal{P}}^2$$

$$\leq \mathbb{E}\|q_k - q\|_{\mathcal{P}}^2 + \mathbb{E}\left\|\eta_k \widehat{\nabla}_q \widehat{\Phi}_N(q_k, u_k)\right\|_{\mathcal{P}}^2 - 2\mathbb{E}\left\langle q_k - q, \eta_k \nabla_q \widehat{\Phi}_N(q_k, u_k) \right\rangle,$$

which implies

$$2\mathbb{E}\left\langle q_k - q, \eta_k \nabla_q \widehat{\Phi}_N(q_k, u_k) \right\rangle \leq \mathbb{E}\|q_k - q\|_{\mathcal{P}}^2 + \mathbb{E}\left\|\eta_k \widehat{\nabla}_q \widehat{\Phi}_N(q_k, u_k)\right\|_{\mathcal{P}}^2 - \mathbb{E}\|q_{k+1} - q\|_{\mathcal{P}}^2$$

for any $q \in \mathcal{P}$. Similarly, we have

$$-2\mathbb{E}\left\langle u_k - u, \eta_k \nabla_u \widehat{\Phi}_N(q_k, u_k) \right\rangle \leq \mathbb{E}\|u_k - u\|_{\mathcal{U}}^2 + \mathbb{E}\left\|\eta_k \widehat{\nabla}_u \widehat{\Phi}_N(q_k, u_k)\right\|_{\mathcal{U}}^2 - \mathbb{E}\|u_{k+1} - u\|_{\mathcal{U}}^2$$

Figure 4: Network architectures for $u$ and $f$.

for any $u \in \mathcal{U}$. By the Lipschitz assumption, we have

$$\mathbb{E}\left\|\eta_k\widehat{\nabla}_q\widehat{\Phi}_N(q_k,u_k)\right\|_{\mathcal{P}}^2 \le (\eta_k)^2 L_N^2, \qquad \text{and} \qquad \mathbb{E}\left\|\eta_k\widehat{\nabla}_u\widehat{\Phi}_N(q_k,u_k)\right\|_{\mathcal{U}}^2 \le (\eta_k)^2 L_N^2.$$

Therefore,

$$2\mathbb{E}\left\langle q_k - q, \eta_k\nabla_q\widehat{\Phi}_N(q_k,u_k)\right\rangle - 2\mathbb{E}\left\langle u_k - u, \eta_k\nabla_u\widehat{\Phi}_N(q_k,u_k)\right\rangle$$
$$\le \mathbb{E}\left(\|q_k - q\|_{\mathcal{P}}^2 - \|q_{k+1} - q\|_{\mathcal{P}}^2 + \|u_k - u\|_{\mathcal{U}}^2 - \|u_{k+1} - u\|_{\mathcal{U}}^2\right) + \left[(\eta_k)^2 + (\eta_k)^2\right]L_N^2.$$

Lastly, by telescoping, we have

$$\mathbb{E}\left\{\sum_{k=0}^{T}\left(2\left\langle q_k - q, \eta_k\nabla_q\widehat{\Phi}_N(q_k,u_k)\right\rangle - 2\left\langle u_k - u, \eta_k\nabla_u\widehat{\Phi}_N(q_k,u_k)\right\rangle\right)\right\}$$

$$\le \mathbb{E}\left(\|q_0 - q\|_{\mathcal{P}}^2 + \|u_0 - u\|_{\mathcal{U}}^2 - \|q_{T+1} - q\|_{\mathcal{P}}^2 - \|u_{T+1} - u\|_{\mathcal{U}}^2\right) + \sum_{k=0}^{T}\left[(\eta_k)^2 + (\eta_k)^2\right]L_N^2$$

$$\le \mathbb{E}\left(\|q_0 - q\|_{\mathcal{P}}^2 + \|u_0 - u\|_{\mathcal{U}}^2\right) + \sum_{k=0}^{T}\left[(\eta_k)^2 + (\eta_k)^2\right]L_N^2$$

$$\le D_{\mathcal{P}}^2 + D_{\mathcal{U}}^2 + \sum_{k=0}^{T}2\eta_k^2 L_N^2.$$

This holds for any $q \in \mathcal{P}$ and $u \in \mathcal{U}$, and therefore holds for the supremum over $q$ and $u$. Hence, we reached the conclusion.

## E   Neural Network Architecture for Biological Dynamic System Experiment

Below, we describe the structure for the neural network we used in the simulation of the ecological dynamical system. The network structure is shown in Figure 4. On the left-hand side, we have the network structure for $f(Y,\xi)$, for which we first input $y_1,\ldots,y_n$ into an RNN comprised of LSTM cells, and then use its output combined with $\xi$ as the input for a fully connected layer. On the right-hand side, the network structure for $u(\theta,Y)$ is similar, except that we replace $\xi$ with $\theta$ and change the dimension of the fully connected layer accordingly. When $\theta$ is generated from $f(Y,\xi)$, we concatenate the two neural networks, using the output of $f$ as part of the input to $u$.

## Footnotes

[6] Imagine Gaussian mixture representation with bounded coefficients, or neural networks whose last layer of activation is a bounded function, such as the hyperbolic tangent or the sigmoid function, amplified by a constant $C_u$ such that $|u(\theta, y)| \leq C_u$.

[7] For example, for $\chi^2$-divergence, we have $C_\nu = C_u^2 + C_u$.

[8] For transportation reparametrization with neural networks, it is known that when the neural network is feed forward and piecewise linear, there exists upper bounds for its pseudo dimension [Bartlett et al., 2017].