[Reviews · NeurIPS 2018]

Reviewer 1



The paper proposes a framework to work with approximate Bayesian computation in an optimisation setting. While the idea is interesting, the presentation is a bit limited: there are many English errors and there is a clear point which is not well explained: from equation (5) on the loss function is intended to be either negative or positive, so that the minimisation problem is meaningless.

Reviewer 2



Thankyou for your detailed response. I am happy with all of your responses, though slightly confused over Q2 (rev2). One can't draw samples from improper priors in the first place, and other techniques (such as Rodrigues et al) won't save you there. You simply need to draw your samples from a distribution that is not the prior. I am still positively inclined towards this paper, and following the response and comparison to EP-ABC I will increase my score to 7 (from 6). -------- The paper presents all ABC algorithms, including the one proposed here, as drawing samples from the prior \pi(\theta). Of course when the prior is improper or merely diffuse with respect to the posterior this will be impossible or at best highly inefficient. So people tend to instead sample from p(y|\theta)g(\theta) for some g that can place samples in regions of high posterior density (perhaps after a quick pilot analysis) and then reweight the samples appropriately. So my question is how P-ABC performs when the prior is improper or diffuse \pi(\theta)\propto 1, say. Theorem 1 works when y_1^*,\ldots,y_n^* are iid (which is curiously why it was written this way on in equation (2)), but presumably not otherwise. While I am glad to see the non-iid analysis in section 4.2 (and so perhaps you should remove the iid statement in (2) to make it more general and realistic as ABC analyses don’t usually use iid data - the non-trivial ones at least) - I wonder what you can say along the lines of Theorem 1 in the non-iid case? Otherwise we have to trust to luck(!) in the more challenging ABC cases, and there are of course more ABC algorithms that work well if the data are iid draws from the model (e.g. Barthelme’s EP-ABC in JASA, for example). Following on from this last point, your toy example involves iid data, and so there are ABC algorithms that work very well in this setting, such as EP-ABC which does not involve summary statistics and scales reasonably well to high dimensions in the number of parameters and very well in the number of datapoints. So I’m wondering if you’re making the right comparisons with your choice of competitor algorithms? Overall I would have liked to have seen the algorithm’s performance in a couple of more situations rather than a simple toy model and then 1 real example to better judge it’s capabilities. When will the method be valuable to use? When will it not work as well as other approaches? etc. I support the motivation to extend ABC to high-dimensions, but the paper seems to have missed a few other methods designed to do this (or designed to improve standard ABC so that it can be extended to higher dimensions). These include Rodrigues et al's recalibration adjustment, Nott et al 2014's marginal adjustment (both of these utilised as regression-type-adjustments) and Li et al 2017's copula ABC (which makes a copula approximation of the posterior). You comment on p.2. para 3 that (1) is a more general case than the synthetic likelihood, and so can have better accuracy. Generally this could be true, though if the distribution of the summary statistics is indeed normal (or approximately normal), then you will do far better with the synthetic likelihood for sure. Finally, please define all acronyms at the first usage (this is mostly done, but not all of the time), capitalise appropriately in the references section, check for spelling/typos/missing-brackets, and tell me what the word “modelstive” means, as it sounds excellent.

Reviewer 3



Thank you for your response that answers my main concerns. I am still voting for accepting the paper. -------------------------------------------------------------------------------------------- This paper considers the problem of performing Bayesian inference when dealing with intractable likelihood. Approximate Bayesian Computation (ABC) is now a well established method to handle complex likelihood which does not admit closed form or is computationally too expensive to be computed. Nevertheless, such methods present limitations like : - they are delicate to tune as they require a non one-to-one function of the data (summary statistics) and the definition of a neighbourhood of the data with respect to some metric, - they may be difficult to scale up. To overcome such issues, the authors propose to move from the initial ABC paradigm towards a saddle point formulation of the problem. They suggest to use the ability to sample from complex model (like in ABC) in order to solve an optimisation problem which provides an approximation of the posterior of interest. The authors compare the performance of their approach with some particular examples of ABC algorithms on synthetic and real datasets. The motivation of the paper and the advocated solution are well explained. The existing literature on ABC is rather dense but the authors managed to give a broad overview of the different solutions. I have only few comments about the existing literature that is not mentioned in the paper: - to be completely fair, the authors could have mentioned ABC random forests for Bayesian parameter inference (Raynal et al. 2016) as it is a solution which avoids tuning the neighbourhood and reduces the need for summary statistic selection, - regarding the formulation in terms of optimisation problem, it would have been interesting to provide a discussion, a comparison or a mention of the work of Moreno, et al. (2016, Automatic Variational ABC) and Fasiolo, et al. (2017, An Extended Empirical Saddlepoint Approximation for Intractable Likelihoods). Do you have any input or comment on this? The paper is overall properly written and easily to follow. I just have few minor comments about the readability: - the definition of the representation of $\theta$ for the posterior distribution in section section 3.1 (l.199 - l.204) is redundant as it is already defined on line l.97 - l.104. I would suggest to do the mention from l.199 - l.204 straight before section 2.1 (sticking to the word representation rather than reparametrization) as it is used in Equations (4), (5) and so on... and remove it from section 3.1, - Equation (7) and equation (10) are the same. In section 3.1, the authors refer alternatively to (10) or (7). I would suggest to remove equation (10) and introduce the definition of $\Phi$ in equation (7) and then specifying $f(Y,\i)$ and $u(\theta,Y)$ when needed. The idea is original and very interesting as it brings some efficient machine learning techniques where existing ABC methods may fail or struggle. The results seem promising though it would have been interesting to have a more comprehensive comparison with existing methods. The main limitation I see to this approach is the inability to reach the optimal solution of the saddle point formulation (7) as in some variational approaches. This would end up to rough or wrong approximations of $f*$ and $u*$ and hence to a misspecified posterior approximations. Do you have any comments on this? I spotted few typos that I listed below : l.101 modelstive? l.102 mdoels -> models l.108 p(Y|\theta)p(\theta) -> p(Y|\theta)\pi(\theta) l.112 Shouldn't be the expectation of equation (4) with respect to q(\theta|Y)p(Y) instead of with respect to p(\theta|Y)p(Y)? l.134 experessive -> expressive